# Nano-org, a functional resource for single-molecule localisation microscopy data

S. Shirgill[1], D. J. Nieves [1], J. A. Pike [2,3,4], M. A. Ahmed[3], H. Abbott[3], M. H. H. Baragilly [1,5], K. Savoye [1,6], J. D. Worboys [7], K. S. Hazime [8], E. Bruggeman[9], A. Garcia[3], D. J. Williamson [10], P. Rubin-Delanchy[11], R. Peters[12], D. M. Davis [8], R. Henriques [13,14], S. F. Lee [9] & D. M. Owen [1] ✉

The nanoscale organisation of proteins plays a key role in diverse cellular processes, including signalling, adhesion, and structural integrity. Single-molecule localisation microscopy (SMLM) is a super-resolution imaging technique that captures the spatial distributions of proteins in cells with nanometre precision, enabling detailed studies of protein clustering and architecture. However, comparing such data across experiments remains challenging due to a lack of curated, functional resources. Here we present a publicly accessible, curated, and functional resource, termed "nano-org", containing SMLM data representing the nanoscale distributions of proteins in cells. Nano-org is searchable by comparing the statistical similarity of the datasets it contains. This unique functionality allows the resource to be used to understand the relationships of nanoscale architectures between proteins, cell types or conditions, supporting the development of the field of spatial nano-omics.

The nanoscale organisation and oligomerisation of proteins are critical processes in fundamental biology. For instance, nanoscale protein clustering is a key mechanism in regulating signal transduction by orchestrating protein-protein interaction rates[1]. Moreover, the organisation of cytoskeletal components, such as the nanoscale architecture of cortical actin, helps define cell mechanical properties[2,3]. Aberrant protein nanoscale organisation has been implicated in diseases, for example, Alzheimer's disease and type II diabetes[4]. Given this importance, there is a need for a resource to allow researchers to compare protein nanoscale organisations.

High-quality community-driven accessible databases/atlases have been transformative across biology in the areas of predictive protein structure, cell phenotyping and large-scale omics, such as NCBI and PDB[5,6]. However, these platforms have yet to be utilised for comparing protein distributions and assemblies - a field termed spatial nano-omics. Single-molecule localisation microscopy (SMLM) is a fluorescence microscopy technique that provides coordinates of protein distributions in cells with nanometre precision[7]. While SMLM databases exist, they lack two features required to enable spatial nano-omics, curation and functionality[8]; features that are

[1]School of Infection, Inflammation and Immunology, School of Mathematics, Centre of Membrane Proteins and Receptors (COMPARE), University of Birmingham, Birmingham, UK. [2]Institute for Interdisciplinary Data Science and AI, University of Birmingham, Birmingham, UK. [3]Research Software Group, Advanced Research Computing, University of Birmingham, Birmingham, UK. [4]The Research Software and Analytics Group, University of Exeter, Exeter, UK. [5]Department of Mathematics, Insurance and Applies Statistics, Helwan University, Helwan, Egypt. [6]School of Physics and Astronomy, University of Birmingham, Birmingham, UK. [7]Lydia Becker Institute of Immunology and Inflammation, Faculty of Biology, Medicine and Health, Manchester Academic Health Science Centre, University of Manchester, Manchester, UK. [8]Department of Life Sciences, Imperial College London, London, UK. [9]Yusuf Hamied Department of Chemistry, University of Cambridge, Cambridge, UK. [10]Department of Infectious Diseases, School of Immunology and Microbial Sciences, King's College London, London, UK. [11]School of Mathematics, University of Edinburgh, Edinburgh, UK. [12]School of Mathematical and Physical Sciences, University of Sheffield, Sheffield, UK. [13]Instituto de Tecnologia Química e Biológica António Xavier, Universidade Nova de Lisboa, Lisboa, Portugal. [14]UCL Laboratory for Molecular Cell Biology, University College London, London, UK. ✉e-mail: d.owen@bham.ac.uk

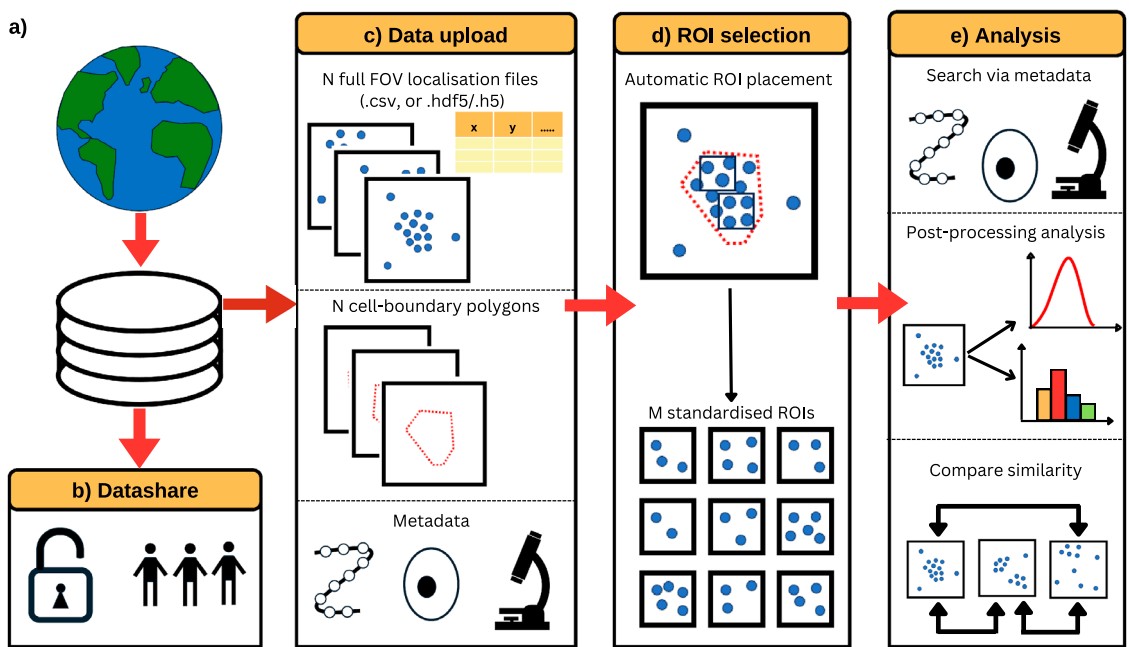

**Fig. 1 | Key functionalities of nano-org. a** Users can upload their data, **b** selecting from various privacy settings. **c** Users can upload localisation files (.csv, or .hdf5/.h5) along with cell-bounding polygons. Metadata requirements ensure comprehensive dataset documentation and enhance search functionality. **d** Uploaded data is split into regions of interest (ROIs) for downstream analysis. **e** The database enables users to explore public datasets, extract relevant information, and utilise statistical similarity tools for comparative analysis.

required to turn a repository into a resource. To address this need, we present nano-org, a publicly accessible and curated database for SMLM data.

Nano-org is a publicly accessible, curated resource for SMLM data that enables the comparison of protein nanoscale organisation across different experimental conditions. It includes features such as metadata curation, real-time updates, and the ability to search datasets by statistical similarity, supporting the study of spatial nano-omics and protein distribution in cells.

## Results

### Data upload, organisation, and metadata curation

Nano-org is freely accessible at nano-org.bham.ac.uk. Uploading and downloading data requires registration and email verification. When processing files containing the coordinates of SMLM localisations, it uses standardised tiled $3 \times 3\,\mu m$ regions-of-interest (ROIs), facilitating the application of the similarity algorithm implemented in the resource. These are produced automatically from full field-of-view data with cell bounding polygons. Datasets are organised into folders representing experimental conditions. They are accompanied by metadata such as the SMLM modality, protein identity, cell type, fluorophore tag and other relevant information (e.g. drug treatments), including a DOI link for published data. The metadata includes whether the data has undergone drift and multiple-blink correction, and also calculates the average localisation precision across datasets, providing users with a quick assessment of data quality. Additionally, upon upload, users are encouraged to include links to raw data files when possible, as well as a correspondence email. This ensures that, where available, raw data can be directly accessed via a link or, alternatively, requested from the uploader, facilitating both data quality verification and collaboration between researchers. To ensure the validity of downstream data analysis, data stored on nano-org is curated and subject to restriction. It currently accepts datasets in .csv or .hdf5/.h5 format acquired from PALM, dSTORM, PAINT, or other SMLM modalities, such as MINFLUX, as specified by the user upon upload. For later analysis, nano-org automatically assesses the localisation density

and coverage. Nano-org then allows users to navigate through the publicly accessible datasets using metadata tags and download pertinent data for their own analysis (Fig. 1).

### Statistical similarity search and ranking of datasets

A key feature of nano-org is the ability to search its contents by the statistical similarity of datasets (Fig. 2). This means users can upload a condition and search for other conditions where proteins exhibit the most similar nanoscale organisation. This is analogous to searching a gene sequence database based on sequence homology. Briefly, on upload, ROIs are subdivided into 30 nm$^2$ bins (a fixed bin size chosen based on the typical localisation precision observed in SMLM experiments, ensuring consistency across datasets). We then form a frequency histogram for the number of localisations in each bin and construct the empirical cumulative distribution function (CDF) of the set of frequencies (histogram heights). Every 5 minutes, a check is run that first identifies all new or modified data in the database. Then, for every pair of ROIs, we compute the largest discrepancy between their CDFs, as in the Kolmogorov-Smirnov (K-S) test[9,10]. The K-S test yields a dissimilarity value ($\lambda$), where an increasing $\lambda$ denotes greater dissimilarity between the ROIs. For condition-wide comparisons the mean dissimilarity, $\bar{\lambda}$, is computed. This procedure results in a list of all other database contents ranked by their nanoscale organisational similarity to the condition in question. This list can then be downloaded and further filtered or searched using metadata. Two different versions of this ranked list are produced – one as described above and one which aims to be invariant to differences in the total number of localisations in the ROI (e.g. if the user has not controlled for expression level, acquisition time, etc). This is achieved by thinning uploaded datasets to a standard value (100 localisations/$\mu m^2$); the minimum density at which the algorithm could identify meaningful differences between datasets while also minimising the exclusion of sparse datasets (see Methods section for detail on how thinning is done). The system scales efficiently with the number of ROIs in the database, as it compares precomputed histograms rather than raw data, enabling many comparisons within a reasonable timeframe. The infrastructure supports real-

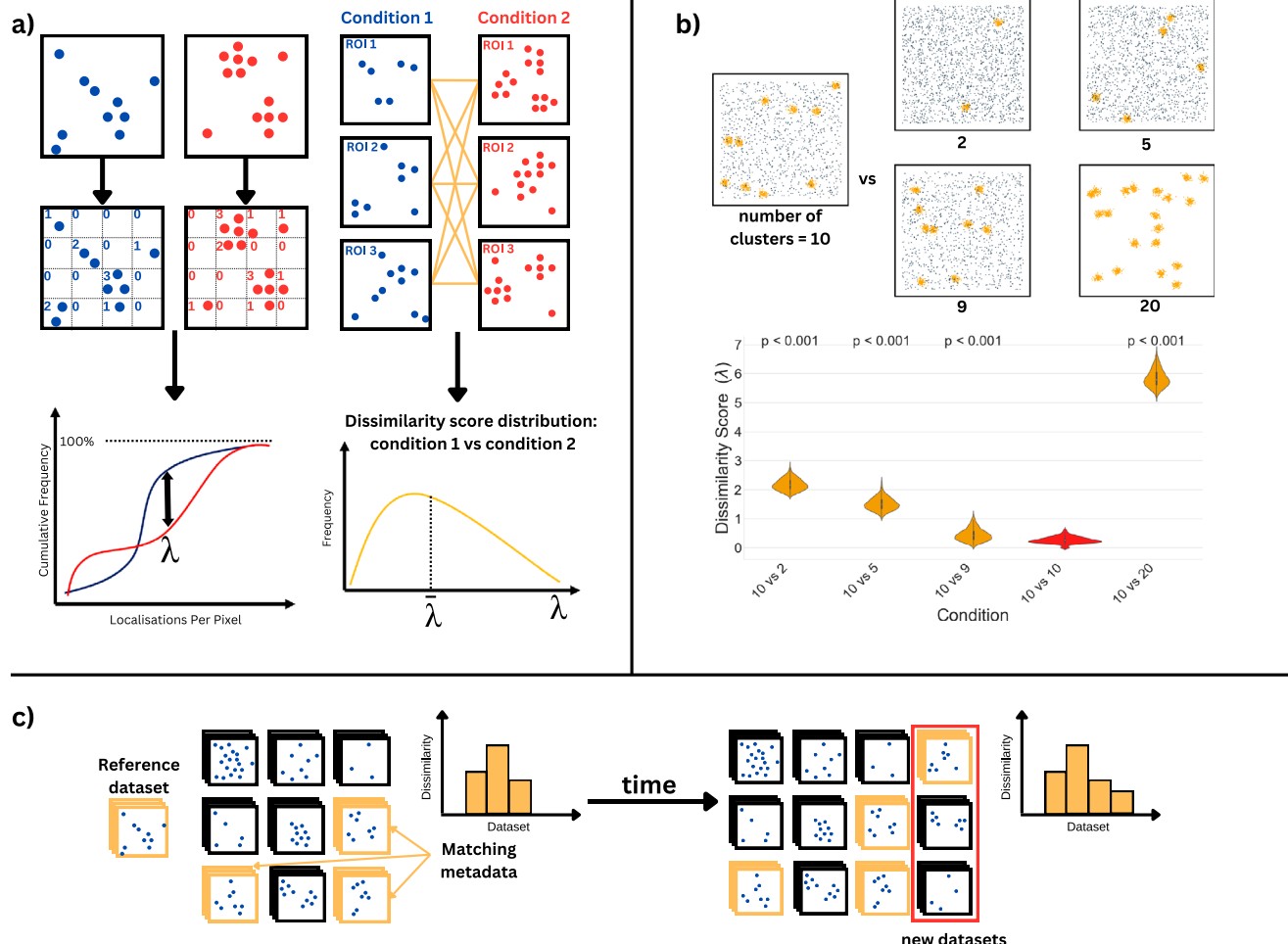

**Fig. 2 | Nano-org's similarity search approach. a** Two ROIs are divided into 30 nm² bins to generate cumulative frequency histograms of localisations. A K-S test yields a dissimilarity value (λ) between the ROIs, with mean dissimilarity (λ̄) calculated across datasets for comparison. **b** Example 3 × 3 μm simulated data regions showing increasing dissimilarity as the number of clusters varies from 10 (30 ROIs per condition), while keeping the total localisations per ROI constant. **c** Real-time analysis on nano-org enables continuous comparison with public datasets, filterable by metadata. Ranked similarity lists are updated with new uploads.

time updates and comparisons, ensuring timely results even as the database grows. A complete description of the algorithm is provided in the Methods section.

### Validation of similarity scoring using simulated data

Our similarity scoring method underwent rigorous testing on simulated data. For example, ROIs with 10 clusters were compared to ROIs with different numbers of clusters while keeping the total number of localisations the same. Dissimilarity increased with the difference in the number of clusters (Fig. 2), and statistical testing demonstrated that the differences were statistically significant (See the Methods for a detailed explanation of the significance testing procedure). Testing with different cluster sizes and varying numbers of points per cluster is shown in Supplementary Fig. S1. Additionally, Supplementary Fig. S2 presents dissimilarity scores for alternative structures, including mixtures of Gaussian-clustered and fibrous localisations, variations in fibre density, and different cluster shapes. Finally, simulated fibrous data with different spatial arrangements is shown in Supplementary Fig. S3.

### Impact of dataset heterogeneity on similarity scores

Supplementary Fig. S4 further explores the effect of data heterogeneity on similarity scores. When datasets contain a fixed number of clusters, self-similarity scores (i.e. the similarity between ROIs within the same dataset) remain tightly distributed. However, as variability

increases (either due to differences in the number of clusters per ROI or the presence of distinct subpopulations) self-similarity scores become broader and can exhibit multiple peaks. This highlights the importance of considering dataset heterogeneity when interpreting similarity scores, as multiple subpopulations within a condition can influence overall comparisons.

### TIGIT organisation in immune cells

To illustrate the utility of our approach, we investigated the organisation of one of the datasets stored on the database – T-cell immunoreceptor with immunoglobulin and ITIM domains (TIGIT); imaged using dSTORM. TIGIT is an inhibitory receptor on various immune cells, including T and NK cells. Recent findings show that upon ligation, TIGIT forms nanoclusters co-localised with the activating T cell receptor, and this clustering is important for its signal transduction[11]. From the ranked similarity list, we found that TIGIT organisation in NK cells was more similar to TIGIT in other cell types, specifically CD4+ T cells than it is to other proteins, such as KIR2DL1 and NKp30, on the surface of NK cells (Fig. 3). This suggests that the protein identity, rather than cell type, is most important in defining the nanoscale organisation in this case. Rankings are preserved after thinning the data showing the trends are due to genuine differences in the protein nanoscale distribution and not solely due to differences in expression levels. It is recommended to use thinned

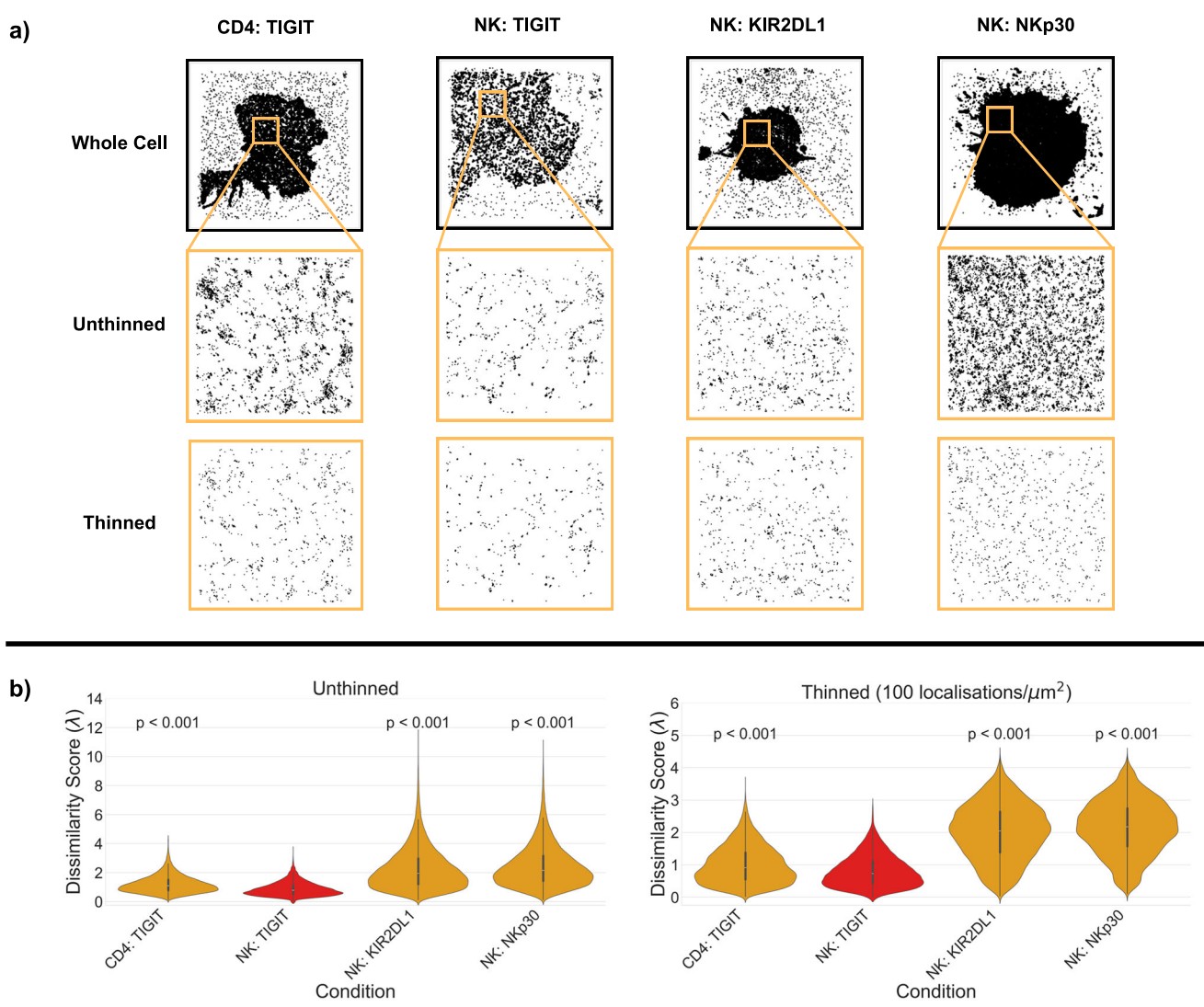

**Fig. 3 | Dissimilarity scores between experimental data in nano-org. a** Examples of whole cell coordinates used for analysis, along with example ROIs for each condition of interest: TIGIT in CD4+ T cells (116 ROIs), and NK cells (158 ROIs); NKp30 in NK cells (416 ROIs); and KIR2DL1 in NK cells (202 ROIs). 3 × 3 μm ROIs are presented either unthinned or thinned to 100 localisations/μm². **b** Dissimilarity between TIGIT in NK cells with itself (red) and with all other conditions.

dissimilarity scores when comparing datasets with significantly different localisation densities. The standard thinning density of 100 localisations/μm² was selected as a balance between sensitivity and inclusivity. As shown in Fig. S5, the ability of the similarity metric to distinguish between different clustering patterns diminishes as the density of localisations decreases. At very low densities (e.g. 10 localisations/μm²), differences between structurally distinct patterns become less pronounced, reflecting a reduced sensitivity of the algorithm under sparse sampling conditions. Conversely, applying a very high thinning threshold may exclude datasets with naturally lower expression levels, limiting the comparability across experimental conditions. Thus, 100 localisations/μm² was chosen to retain discriminatory power while ensuring that datasets with moderate localisation densities remain tractable.

### Robustness of similarity scores to technical variability
Additionally, further testing on experimental datasets demonstrated that while dissimilarity scores can capture subtle differences introduced by image processing methods, most fitting algorithms produced reconstructions that were visually and structurally similar to the reference (Fig. S6a). While some differences were observed when comparing datasets acquired on different imaging platforms (e.g. ONI

vs. N-STORM), likely due to technical variations such as laser power or detector sensitivity, these were relatively minor compared to the dissimilarity observed when comparing biologically distinct samples imaged in different labs (Fig. S6b). This highlights that, although microscope settings can introduce subtle variability, the dissimilarity scores are more strongly influenced by underlying biological structure.

### Effect of drug treatment on nanoscale organisation
Furthermore, when comparing cells treated with increasing doses of nocodazole, a drug that disrupts microtubules, dissimilarity scores reflected the expected changes in microtubule organisation (Fig. S6c). Cells treated with 0.5 μg/ml nocodazole showed moderate differences from untreated controls, while those treated with 5 μg/ml displayed markedly greater dissimilarity, consistent with a loss of filament structure.

## Discussion
In conclusion, nano-org is a publicly accessible and curated database of SMLM data, designed to facilitate collaborative data sharing, enhancing accessibility and reproducibility. Its unique framework allows searches based on statistical similarity, enabling investigations into the biophysical mechanisms of nanoscale organisation and the

effects of mutations or treatments on protein distributions. This resource contributes to the development of spatial nano-omics – the systematic study of cellular nanoscale architecture.

## Methods

### Nano-org: technical description

Nano-org is a Django-based website, with an SQLite database backend, which is hosted on a BEARCloud virtual machine at the University of Birmingham. Celery and RabbitMQ are used to schedule background tasks, including checking the database for new and modified data and initiating computationally intense analysis tasks via job submission to BlueBEAR – the University of Birmingham's supercomputer for high-performance computing (HPC). Celery tasks are also used to retrieve analysis results from HPC jobs and incorporate them into the website and database. Uploaded data and analysis results are stored on the University of Birmingham's central Research Data Store and made available to download through the website. Core analysis functionality, including cumulative histogram generation and Kolmogorov-Smirnov (K-S) score calculation, is incorporated into our stand-alone Python package smlm-analysis. This package is utilised by nano-org but can also be used by researchers who want to develop their own customised analysis pipelines.

### Dissimilarity algorithm

To compare the dissimilarity between two ROIs, let F(x) and G(x) be their empirical CDFs with sample sizes m and n, respectively. Here, the sample sizes are the number of $30 \times 30$ nm bins within the ROI that contain at least one localisation. A two-sample K-S statistic is employed using the definition,

$$D_{mn} > c(\alpha)\sqrt{J}, \tag{1}$$

where,

$$D_{mn} = \max_x |F(x) - G(x)|, \tag{2}$$

$$J = \frac{n+m}{nm}, \tag{3}$$

$$c(\alpha) = \sqrt{-\frac{1}{2}\ln\left(\frac{\alpha}{2}\right)}. \tag{4}$$

Here, $D_{mn}$ is the maximum difference between F(x) and G(x), which are the CDFs of the two ROIs. J adjusts for sample sizes as the K-S test is sensitive to dataset size. The parameter $\alpha$ is the confidence level, where we set $\alpha = 0.05$. We define,

$$\lambda = \frac{D_{mn}}{c(\alpha)\sqrt{J}} \tag{5}$$

Where $\lambda$ is then the dissimilarity score between the two datasets, ensuring that comparisons remain consistent regardless of differences in the number of localisations. More details are provided in[12].

A value of $\lambda = 0$ indicates that the two ROIs have identical distributions, while higher $\lambda$ values indicate increasing dissimilarity. A score greater than 1 ($\lambda > 1$) indicates that two datasets are significantly different. However, a score less than 1 ($\lambda < 1$) does not necessarily imply that the datasets are highly similar—it simply means that their differences are not statistically significant. If comparing similarity scores across multiple datasets to establish a ranking of similarity, while some scores may fall below 1, the relative ordering of scores remains informative, helping to rank datasets by their degree of similarity or dissimilarity.

A caveat of the dissimilarity algorithm is that results are influenced by localisation density, i.e. datasets with lower densities tend to have lower dissimilarity scores. To obtain dissimilarity scores that are independent of localisation density, we perform thinning before computing dissimilarity scores. The thinning process involves:

- ROI processing: For each file in a dataset, $3 \times 3\,\mu m^2$ ROIs are extracted.
- Random subsampling: Each ROI within the entire dataset is randomly subsampled 100 times to achieve a density of 100 localisations/$\mu m^2$ per ROI.
- Handling sparse datasets: If a ROI has a density fewer than 100 localisations/$\mu m^2$, it is excluded from the analysis to prevent bias.
- Thinned histogram generation: Frequency histograms are generated for each subsample. The histograms are then averaged across all 100 repeats to ensure robustness.
- Dissimilarity computation: For each ROI in the reference condition, the dissimilarity score is computed against ROIs in the comparison condition, and the mean dissimilarity score along with its standard deviation is reported. This normalisation allows for density-independent comparisons while preserving the underlying protein distribution patterns.

It is recommended to use thinned dissimilarity scores when comparing datasets with significantly different localisation densities.

### Data simulations

To model protein distributions that exhibit clustering behaviour, we generate Gaussian-distributed clusters within a $3 \times 3\,\mu m^2$ ROI. Each simulation follows these steps:

1. Cluster Generation: A specified number of clusters (n) are randomly positioned within the ROI.
2. Point Distribution: Each cluster contains p localisations, which are sampled from a Gaussian distribution centred at the cluster position.
3. Cluster Variability: The spread of each cluster is controlled by the standard deviation ($\sigma$), determining the tightness of clustering.

This approach ensures that clusters of varying sizes and densities can be systematically compared using the dissimilarity algorithm.

To simulate the nanoscale organisation of cytoskeletal-like structures, we generate linear fibre distributions within an ROI. The fibre generation process involves:

1. Defining Fibre Orientation: Fibres can be randomly oriented or aligned parallel within the ROI.
2. Fibre Placement: The number of fibres is specified by the user, and their positions are randomly or uniformly distributed.
3. Point Distribution Along Fibres: Localisations are assigned along each fibre's length following a linear pattern, mimicking filamentous structures such as cytoskeletal networks.

This method captures the spatial arrangement of fibrous networks, enabling systematic comparisons between ordered and disordered fibre architectures.

For more details of how simulations are generated, see https://gitlab.bham.ac.uk/owendz-protein-databank/nano-org-similarity-scoring.

### Experimental methods

All experimental datasets, with the exception of the microtubule data, were obtained from collaborating laboratories. Microtubule data were acquired in-house under the conditions described below. All datasets used in this study are publicly available via nano-org.

## Cell culture

COS-7 cells were maintained in Dulbecco's Modified Eagle Medium (DMEM, high glucose; Sigma-Aldrich), supplemented with 10% fetal bovine serum (FBS; Gibco, Life Technologies), 1% penicillin/streptomycin (Gibco, Life Technologies), and 1% L-glutamine (Gibco, Life Technologies), at 37 °C in a humidified incubator with 5% $CO_2$. For imaging, cells were seeded at a density of $10^4$ cells per well in an eight-well μ-slide (Ibidi, high precision glass bottom) one day prior to fixation.

For experiments involving nocodazole treatment, cells were incubated for 30 min at 37 °C in DMEM containing either 0.5 μg/mL or 5 μg/mL nocodazole. Following treatment, cells were washed three times with phosphate-buffered saline (PBS). Untreated control cells were washed directly with PBS.

Cells were subsequently subjected to a sequential extraction and fixation protocol. Briefly, cells were extracted with a pre-warmed solution of 0.25% Triton X-100 and 0.1% glutaraldehyde in PEM buffer (80 mM PIPES, 5 mM EGTA, 2 mM $MgCl_2$, pH 6.8) at 37 °C for 90 s. This was followed by fixation in a solution of 0.25% Triton X-100 and 0.5% glutaraldehyde in PEM at 37 °C for 10 min. Post-fixation, samples were quenched with 1 mg/mL sodium borohydride (Sigma-Aldrich) for 7 min[13] and washed three times with PBS.

## Immunolabelling

Cells were permeabilised with 0.1% Triton X-100 in PBS for 3 min at room temperature (RT), cells were then washed in PBS followed by blocking in 5% bovine serum albumin (BSA; Sigma-Aldrich) for 30 min. Immunostaining was performed using a mouse monoclonal β-tubulin $IgG_3$ primary antibody (200 μg/mL; Santa Cruz Biotechnology), diluted 1:50 in 5% BSA and incubated for 30 min at RT. After three PBS washes, cells were incubated in Alexa Fluor™ 647-conjugated goat anti-mouse IgG secondary antibody (2 mg/mL; Life Technologies), diluted 1:1000 in 5% BSA, for 30 min in the dark at RT. Samples were then washed five times with PBS. Prior to dSTORM imaging, PBS was replaced with an imaging buffer consisting of 18% glucose (w/v), 10 mM Tris (pH 8), 50 mM NaCl (Sigma-Aldrich), 0.8 mg/mL glucose oxidase, 50 mM cysteamine (Sigma-Aldrich), and 40 μg/mL catalase (Sigma-Aldrich).

## Optical setup

Microtubule data was collected using an ONI Nanoimager S microscope unless otherwise stated. Where indicated, a Nikon N-STORM microscope was used for comparison.

## Data analysis

Data analysis was conducted using the Super resolution Microscopy Analysis Platform (SMAP)[14], with default settings applied unless stated otherwise. Single-molecule localisations were fitted using the *PSF free* algorithm. To assess the impact of different localisation algorithms on similarity metrics, localisations were also fitted using the *ellipt:PSFx PSFy* or *PSF fix* algorithm in SMAP, as well as several alternative fitting methods available in ThunderSTORM[15], including *Gaussian (Gaus)*, *integrated Gaussian (integrated Gaus)*, *centroid*, and *radial* fitters.

Localisations were filtered to exclude those with an estimated precision greater than 30 nm. Drift correction and grouping were performed to mitigate the effects of sample drift and multiple blinking of fluorophores.

## Summary of statistical testing method

To assess whether dissimilarity scores comparing different conditions to a reference condition were statistically significant, p-values were calculated using Monte Carlo simulations and permutation testing. The analysis began by extracting all dissimilarity values comparing the reference condition with itself and with each specific comparison condition. For the statistical comparison, we conducted 1000 Monte Carlo simulations. A distance matrix was computed using these dissimilarity values. The test statistic, calculated from this distance matrix, quantified the differences between intra-group and inter-group distances. The p-value was determined by comparing the observed test statistic with those obtained from permuted data, representing the proportion of permuted test statistics greater than or equal to the observed statistic. To validate this approach, we repeated the p-value calculation 500 times and plotted the empirical cumulative distribution function (ECDF) of the resulting p-values. This method ensures a robust comparison of group differences and the reliability of the calculated p-values.

## Reporting summary

Further information on research design is available in the Nature Portfolio Reporting Summary linked to this article.

## Data availability

All experimental data is stored and available for download on https://nano-org.bham.ac.uk.

## Code availability

The implementation of the website and database is available at https://gitlab.bham.ac.uk/owendz-protein-databank/nano-org-website[16]. The core analysis functionality and algorithms used by nano-org are implemented as a stand-alone python package which is available at https://gitlab.bham.ac.uk/owendz-protein-databank/smlm-analysis[17]. All Python scripts used to produce simulated data and violin plots in figures and Supplementary Figs. are available at https://gitlab.bham.ac.uk/owendz-protein-databank/nano-org-similarity-scoring[18].

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

## Acknowledgements

The research described in this paper was carried out with the assistance of Advanced Research Computing at the University of Birmingham. This included support from the Research Software Group to develop the database and website, data storage on the Research Data Store, computations on the BlueBEAR HPC service and use of a BEAR Cloud Virtual Machine to host the website. DMO and SS acknowledge funding from the Biotechnology and Biological Sciences Research Council (BBSRC) grant BB/X018644/1. DMD acknowledges funding from the Medical Research Council (MRC) grant MR/W031698/1, supported by a Welcome Career Development Award (307027/Z/23/Z to JDW). RH received funding from the European Research Council (ERC) through grant 101001332-SelfDriving4DSR and Horizon Europe through grants 101057970-AI4LIFE and 101099654-RTSuperES. Views and opinions expressed are, however, those of the authors only and do not necessarily reflect those of the European Union. Neither the European Union nor the granting authority can be held responsible for them. This work was also supported by a European Molecular Biology Organization (EMBO) installation grant (EMBO-2020-IG-4734 to RH).

## Author contributions

S.S., D.J.N., M.H.H.B., K.S., and J.A.P. developed and tested the algorithms. J.A.P., H.A., M.A.A., and A.G. implemented the database and website. D.J.N., J.D.W., K.S.H., D.J.W., R.P., E.B., and D.M.D. provided simulations, experimental data and testing. PRD supported with statistical analysis. R.H., S.F.L., and D.M.O. conceived the work. S.S. and D.M.O. wrote the manuscript.

## Competing interests

S.F.L. is a cofounder of ZOMP. The remaining authors declare no competing interests.
