## [Transparent Peer Review file · Nature Communications]

Nano-org, a functional resource for single-molecule localisation microscopy data

Corresponding Author: Professor Dylan Owen

Version 0:

Reviewer comments:

Reviewer #1

(Remarks to the Author)

Sharing of data across the scientific community, ideally such that it can be reanalyzed is invaluable. The comparison of data acquired in one laboratory with those acquired across the community allows for validation and interpretation. Data sharing thus is invaluable within a community of users. Straightforward access to databases for users that don't use the corresponding approaches can allow such users to complement their own technically different experiments with community data on experiments that they can't or don't have to execute. This can make a database invaluable across the entire scientific community. Examples are AlphaFold for protein structure and the long-established nucleic acid and protein sequence databases. Here Shirgill et al. present a new database for single molecule localization microscopy (SMLM) data. This has to be commended. The two key questions in assessing the utility of this resource are: How well does the data type lend itself to comparison across the community? How easy is it to access the data for reanalysis from within and beyond the user community? To allow the editor and authors to assess the following comments, this reviewer has a substantial background in various imaging approaches but has engaged with dSTROM and PALM only at an exploratory level.

SMLM data are difficult to acquire and prone to technical artifacts (see e.g. doi: 10.1038/s41590-018-0162-7). On one hand this makes a data depository valuable, in giving non-experts access to SMLM data. On the other hand, assessing whether the individual data sets in the database are technically sound has to be taken on trust. This reviewer appreciates that some information on data processing is given with each dataset, yet from the outside this doesn't help in assessing technical quality. It makes sense to present all SMLM data as .csv localization files. However, if the database could provide a link to the raw data, if available (and manageable), that could help the community to comment on each other's data.

Nucleic acids and proteins are reasonably stable on experimental time scales of minutes to hours and can effectively be determined without consideration of subcellular space. This helps comparability, as data can be acquired in standard cell lysates and aren't too dependent on the dynamics of cellular activation. SMLM data are drastically different. Molecular localizations can change on the seconds time scale and are by definition dependent on subcellular localization. This makes the biological interpretation of comparisons between data sets highly context dependent. Are similar parts of a cell compared? Does that happen at similar times in the cellular activation process? In this context, the link to publications to assess details of experimental execution is highly welcome.

The reviewer appreciates that the authors have developed a single number to compare the similarity of molecule distributions between two database entries, the dissimilarity score. Leaving the previous comment on general comparability between samples aside, it still seems difficult to assign biological meaning to the score. For example, at an intermediate score level, are there parts of the two compared distributions that are highly similar (e.g. through joint protein cluster formation) with the rest random or is there a consistently intermediate similarity across the entire sample? A graphical output of the comparison that identifies spatial features of similarity could help. At this point the database produces a ranked list of pairwise dissimilarity scores. Is there any way to extend that to a clustering algorithm across multiple samples to identify groups of similar molecule distributions?

A quick technical question. The data are deposited in 3x3µm tiles. It seems likely that molecule distributions differ between different tiles from the same data set, and such difference likely will carry biological information, e.g. with differences between the center and edge of cellular interfaces. How does the comparison method deal with the potential need to consider multiple 3x3µm tiles per sample?

In summary, contrary to highly successful nucleic acid and protein databases a database of SMLM data is more challenging because of the technical difficulties in acquiring and analyzing the data and the inherent spatial and temporal variability of molecule distributions. This reviewer believes that such database can still be an invaluable resource for the SMLM

community. A reviewer from within this community will have to comment on the severity of the challenges in sample comparability listed here. It is difficult to see how scientists from outside the SMLM community will interact effectively with the database. A clustering feature may be most useful.

(Remarks on code availability)

Reviewer #2

(Remarks to the Author)

Shirgill et al. present a novel database, "Nano-org," designed for sharing data in SMLM. This database has the potential to simplify data comparison between research groups, especially in the context of publishing, while enhancing straightforward replication. However, several aspects of the manuscript require improvement and clarification:

- Alongside details on protein identity and cell type, the experimental localization precision should be reported [Endesfelder et al., *Histometry and Cell Biology*, 2014]. This could be automatically computed upon data submission to the database.

- Data submission should not be limited to 2D data.

- While the .csv format is commonly used in SMLM, it is not the only format. Consider including support for .hdf5 (Picasso) and .mat (SMAP) formats to ensure broader compatibility.

- The three allowed modalities are PALM, dSTORM, and PAINT. Maybe the authors might consider expanding the list to include other relevant methods, such as MINFLUX.

- The algorithm for similarity quantification requires further validation. The current approach appears too simplistic given the diversity of structures imaged in SMLM. Provide quantitative examples comparing datasets of different proteins and cell types. Demonstrate how the algorithm handles datasets of the same sample imaged by different labs. A recurring issue was noted where many datasets appeared most similar to the same dataset (Jurkat_TIGIT), which may suggest a flaw in the algorithm.

- Data is split into bins based on average localization precision (LP). Clarify whether LP is theoretical (per localization) or experimental (one value for the whole dataset). If theoretical, consider switching to experimental values, as theoretical precision may be inaccurate due to camera miscalibration and photon conversion errors.

- The manuscript claims rigorous testing on simulated data but presents only a few examples. Include additional examples using well-characterized biological structures (e.g., NPC). Provide results from multiple labs imaging the same sample for cross-validation.

- In Figure 2b, p-values are reported without an explanation of their calculation. In Figure S2a, the p-value for comparisons between "100 vs. 90" and "100 vs. 100" is reported as <0.001 , which seems implausibly low for distributions that appear visually similar.

- Figure 3 mentions similarity to the Jurkat TIGIT dataset, which, as previously noted, appears excessively similar to multiple other datasets. This raises concerns about the accuracy of the similarity metric.

Overall, the database concept is promising but requires more rigorous validation and expanded data handling capabilities to ensure reliability and broad usability within the SMLM community.

(Remarks on code availability)

Reviewer #3

(Remarks to the Author)

The authors have developed a resource, termed nano-org, that contains curated SMLM data of different proteins in cells. Users can register and upload new SMLM datasets to the database, which includes tags for the cell type, fluorophore, protein and some limited metadata information. The key feature is the "dissimilarity metric" that shows which datasets are most similar to the data that a user is interested in.

As it stands, the resource will be useful to a limited number of researchers interested in clustering of membrane proteins or actin and microtubule filaments and the way the metrics are computed is not broad enough to consider other SMLM data types, such as those of organelles, nuclear structures and others. The following shortcomings should be addressed to make it more robust and more broadly useful.

1. The methodologies used are generally poorly described in the manuscript. For example, the description of the dissimilarity algorithm is extremely short and hard to understand. Similarly, there are no details provided about how the simulations are performed. It would be beneficial to the reader to have a more in-depth description of these methods.

2. Along these lines, the database does thinning to fairly compare data sets with different densities (by for example different measurement conditions). How is this thinning performed? Just a random selection of points? And what happens if the data is sparser than 30 localizations per μm^2 . It is also unclear how this comparison works, as many of the databank entries have multiple files for the entire dataset. Are these first combined into the full structure, or are they analyzed separately?
3. A link on the databank website that points to an “algorithm explanation/help” page would also be useful, so users can double check what these scores mean.
4. 20+ of the datasets (out of 33) are “most similar” with the Jurkat_TIGIT data set, even the “actin HeLa Phalloidin 488” and “actin HeLa Phalloidin 647” point to this dataset, whereas one would expect it to point to each other or another actin data set. Did something go wrong in the selection there?
5. While the databank contains some metadata parameters, it does not go far enough to ensure that the data uploaded is of high quality. Points such as signal to noise ratio of the raw data, background, whether the number of fluorophores active per frame is below the sparseness required for the fitting algorithms used, imaging completeness, under-sampling all affect the final image quality. It would be much more useful if the uploaded data was screened and given a quality score. In the absence of this, the usefulness of the curated data and the comparisons is unclear. This limitation should at the very least be discussed.
6. It appears that the viewer can only see the small field of view per file. Some databank entries are so split up (e.g., limited frames) that there is no structure visible. This is particularly true for many of the Lck mEos3.3 Jukart on anti-CD90 datasets, where many fields of view contain just a few, sparse, random localizations. This makes it hard to judge something quickly without first having to download all the files. It would be nice to be able to see the full, unsplit field of view.
7. It is unclear that splitting the fields to $3 \times 3 \mu\text{m}^2$ to do the “dissimilarity” comparison is useful as it assumes that the protein is uniformly distributed throughout the cell. For proteins that show biased localization to specific organelles or sub-cellular compartments (e.g. peri-nuclear region, nucleus etc...), this information will be lost when the fields are randomly split into small fields of view. In addition, proteins that look the same at nanoscale level may not necessarily be the same when cell-scale or mesoscale information is considered and hence, additional metrics in addition to the “dissimilarity metric” are likely necessary to truly compare different datasets. These points must be considered and addressed to make the resource more broadly useful than just looking at clusters of membrane proteins. Otherwise, the “spatial nano-omics” label is misleading.

(Remarks on code availability)

Reviewer #4

(Remarks to the Author)

(Remarks on code availability)

Version 1:

Reviewer comments:

Reviewer #1

(Remarks to the Author)

The authors have carefully addressed technical concerns - thank you.

(Remarks on code availability)

N/A

Reviewer #2

(Remarks to the Author)

The authors have addressed all my concerns. I recommend publication of the study.

(Remarks on code availability)

Nothing more to add here.

Reviewer #3

(Remarks to the Author)

The authors have done a significant effort in addressing all the comments of the reviewers. Some (mostly minor) concerns remain as noted below but these should be easy to address by the authors and a re-review is not required.

Comments:

- Figure 1 still only mentions .csv files (figure & legend). This should be updated.
 - When thinning is mentioned in the text (e.g., line 90), a reference should be made to the supplementary information where the information on the process can be found.
- The last statement of the SI that mentions that the thinning dissimilarity value is preferred for data with different densities should also be mentioned in the main text as this is very important information.
- Some dissimilarity scores are below 0 (e.g., figure S6), which does not seem possible according to the definition that is provided in the SI? $c(\alpha)$ is a number that will always be positive, J is always positive, and D_{mn} contains the absolute value, so it is also positive. Is this an artifact from the violin plot representation?
 - The following resource cannot be accessed without an account: For more details of how simulations are generated, see <https://gitlab.bham.ac.uk/shirgils/nano-org-similarity-scoring> (lines 90-91 in the SI). This should be made publicly available as it contains pertinent information to how the simulations are performed.
 - Line 109 of the SI does not appear to be shown correctly.

(Remarks on code availability)

Reviewer #4

(Remarks to the Author)

(Remarks on code availability)

NA

Reviewer 1

1. SMLM data are difficult to acquire and prone to technical artifacts (see e.g. doi: 10.1038/s41590-018-0162-7). On one hand this makes a data depository valuable, in giving non-experts access to SMLM data. On the other hand, assessing whether the individual data sets in the database are technically sound has to be taken on trust. This reviewer appreciates that some information on data processing is given with each dataset, yet from the outside this doesn't help in assessing technical quality. It makes sense to present all SMLM data as .csv localization files. However, if the database could provide a link to the raw data, if available (and manageable), that could help the community to comment on each other's data.
 - a. We thank the reviewer for their comment. In response to this feedback, we have now included a field that allows users to provide a link to access raw data, where available.
 - b. However, we recognise that many researchers store SMLM raw data on local hard drives or institution-specific storage systems, making it impractical to provide a direct link in many cases. To address this, we have also introduced a field encouraging users to include a contact email, allowing interested researchers to request raw data directly. This approach facilitates community engagement, data transparency, and potential collaborations.
2. Nucleic acids and proteins are reasonably stable on experimental time scales of minutes to hours and can effectively be determined without consideration of subcellular space. This helps comparability, as data can be acquired in standard cell lysates and aren't too dependent on the dynamics of cellular activation. SMLM data are drastically different. Molecular localizations can change on the seconds time scale and are by definition dependent on subcellular localization. This makes the biological interpretation of comparisons between data sets highly context dependent. Are similar parts of a cell compared? Does that happen at similar times in the cellular activation process? In this context, the link to publications to assess details of experimental execution is highly welcome.
 - a. To ensure clarity, we have now implemented a metadata field that specifies whether a sample is fixed or live. At present, live-cell SMLM data cannot be uploaded to the database, as analysing dynamic molecular localisations requires different computational approaches than those currently implemented on the platform. However, we recognise the value of live-cell SMLM data and believe that a dedicated database for live-cell imaging, such as single-particle tracking, could be a valuable future project.
 - b. Additionally, we have updated the experimental notes section to encourage users to include a link to published methodologies where available, further supporting the biological context of each dataset.
3. The reviewer appreciates that the authors have developed a single number to compare the similarity of molecule distributions between two database entries, the dissimilarity score. Leaving the previous comment on general comparability between samples aside, it still seems difficult to assign biological meaning to the score. For example, at an intermediate score level, are there parts of the two compared distributions that are highly similar (e.g. through joint protein cluster formation) with the rest random or is there a consistently intermediate similarity across the entire sample? A graphical output of the comparison that identifies spatial features of similarity could help.
 - a. We appreciate the reviewer's comment and agree that a single dissimilarity score may not always provide a complete picture of dataset comparability. To address this, we have now implemented additional visualisations to better

- illustrate the heterogeneity and spatial similarity patterns within and between datasets.
- b. Specifically, we have introduced a histogram of self-similarity scores, which displays all similarity scores for every $3 \times 3 \mu\text{m}^2$ gridded ROI compared against all other ROIs within the same condition. This allows users to assess dataset heterogeneity—for example, if a dataset contains two distinct subpopulations, this will be reflected as two peaks in the histogram.
4. At this point the database produces a ranked list of pairwise dissimilarity scores. Is there any way to extend that to a clustering algorithm across multiple samples to identify groups of similar molecule distributions?
 - a. While we do not recommend using dissimilarity scores as the sole basis for clustering analysis, they serve as a useful exploratory tool for hypothesis generation. The scores provide an initial indication of similarity between datasets, but more detailed analyses, such as clustering, should be performed separately to account for underlying biological complexity.
 - b. To support users in conducting further analysis, for future work we will include a clustering feature on the site. This allows users to explore patterns of similarity across multiple datasets in a more structured way, complementing the insights provided by the pairwise dissimilarity scores.
 5. A quick technical question. The data are deposited in $3 \times 3 \mu\text{m}$ tiles. It seems likely that molecule distributions differ between different tiles from the same data set, and such difference likely will carry biological information, e.g. with differences between the center and edge of cellular interfaces. How does the comparison method deal with the potential need to consider multiple $3 \times 3 \mu\text{m}$ tiles per sample?
 - a. To address this, and in response to a previous comment regarding graphical outputs, we have now included histograms of self-similarity scores. These histograms provide a more detailed view of dataset heterogeneity, revealing whether molecule distributions vary significantly across different $3 \times 3 \mu\text{m}^2$ tiles—for example, through broad distributions or the presence of multiple peaks, which may indicate distinct subpopulations.
 - b. Additionally, the comparison method inherently accounts for multiple $3 \times 3 \mu\text{m}^2$ tiles by computing a similarity score for each tile-to-tile comparison. The final dissimilarity score is then derived by taking the average and standard deviation of all computed comparisons, ensuring that spatial variations within a dataset are reflected in the results.

Reviewer 2

1. Alongside details on protein identity and cell type, the experimental localization precision should be reported [Endesfelder et al., *Histometry and Cell Biology*, 2014]. This could be automatically computed upon data submission to the database.
 - a. We thank the reviewer for their suggestion. In response, we have implemented an automatic calculation of average localisation precision for each dataset upon submission to the database. This value is now displayed alongside metadata such as protein identity and cell type.
2. Data submission should not be limited to 2D data.
 - a. We have expanded the platform to accept datasets beyond 2D data, ensuring broader compatibility with different SMLM formats. However, it is important to note that the current algorithms are designed to process only x,y coordinates

- at this stage, meaning that all analyses on the site are based on a 2D interpretation of the data.
- b.** Expanding the algorithm to incorporate 3D data by utilising the z-coordinate is a valuable future direction with the potential to enhance the platform's capabilities. However, implementing this would require significant algorithmic adaptations and computational optimisations, making it a long-term development goal rather than a near-term update.
 3. While the .csv format is commonly used in SMLM, it is not the only format. Consider including support for .hdf5 (Picasso) and .mat (SMAP) formats to ensure broader compatibility.
 - a.** We appreciate the reviewer's suggestion. In response, we have expanded the range of supported upload formats to include .hdf5 (Picasso) and .mat (SMAP) files, ensuring broader compatibility with commonly used SMLM data formats.
 - b.** To maintain accessibility for all users, datasets downloaded from the platform will be provided in .csv format, ensuring consistency and ease of use across different analysis workflows.
 4. The three allowed modalities are PALM, dSTORM, and PAINT. Maybe the authors might consider expanding the list to include other relevant methods, such as MINFLUX.
 - a.** We thank the reviewer for highlighting this aspect of the work. While the website primarily highlights PALM, dSTORM, and PAINT, it does not restrict users to these three modalities. Upon upload, users have the option to select "Other" under "Modality", allowing datasets from techniques such as MINFLUX and STORM to be included.
 - b.** However, we recognise that the wording in the manuscript may have implied that only these three modalities are accepted. We have now revised the text to clarify that the platform supports a broader range of super-resolution imaging techniques.
 5. The algorithm for similarity quantification requires further validation. The current approach appears too simplistic given the diversity of structures imaged in SMLM. Provide quantitative examples comparing datasets of different proteins and cell types. Demonstrate how the algorithm handles datasets of the same sample imaged by different labs. A recurring issue was noted where many datasets appeared most similar to the same dataset (Jurkat_TIGIT), which may suggest a flaw in the algorithm.
 - a.** A known limitation of the algorithm is that similarity scores can be skewed by localisation density with lower-density datasets tending to yield lower scores. Since the 'Jurkat_TIGIT' dataset has a very low average density, it appears as the most similar dataset to many others when using the unthinned similarity score.
 - b.** To mitigate this bias, the thinned similarity score is computed by equalising localisation densities before applying the algorithm, providing a density-independent comparison. When using thinned scores, 'Jurkat_TIGIT' is no longer the most similar dataset for the majority of cases.
 - c.** To ensure clarity, we have now included guidance in the help section, recommending the use of thinned similarity scores when comparing datasets with significantly different densities.
 - d.** Regarding the reviewer's request for additional quantitative comparisons, we have expanded the supplementary material (Figure S6) to include a broader set of comparisons between datasets from different proteins and cell types, as well

as those acquired by different microscopes, different localisation algorithms and different drug treatments. This is in addition to Figure 3 in the main manuscript. These additions provide further validation of the algorithm across a wider range of experimental conditions and demonstrate its robustness in practical use.

6. Data is split into bins based on average localization precision (LP). Clarify whether LP is theoretical (per localization) or experimental (one value for the whole dataset). If theoretical, consider switching to experimental values, as theoretical precision may be inaccurate due to camera miscalibration and photon conversion errors.
 - a. We thank the reviewer for their comment, and we appreciate the opportunity to clarify this point. The manuscript previously lacked a clear explanation of how bin size is determined. To clarify, bin size is not calculated individually based on LP for each dataset. Instead, it is fixed at 30 nm² for all datasets, as this order of magnitude represents a typical LP observed in SMLM experiments.
 - b. We have now updated the manuscript to explicitly state this, ensuring that the methodology is clearly communicated.
7. The manuscript claims rigorous testing on simulated data but presents only a few examples. Include additional examples using well-characterized biological structures (e.g., NPC). Provide results from multiple labs imaging the same sample for cross-validation.
 - a. We thank the reviewer for their recommendation. In response, we have expanded our testing on simulated data to include a broader range of cluster shapes and structural variations, providing a more comprehensive evaluation of the similarity metric. Additionally, we have incorporated cross-validation experiments where the same sample was imaged in different labs with different microscopes, allowing us to assess the robustness and reproducibility of the approach across different experimental conditions. These results have been added to the Supplementary Material (figure S2).
8. In Figure 2b, p-values are reported without an explanation of their calculation. In Figure S2a, the p-value for comparisons between "100 vs. 90" and "100 vs. 100" is reported as <0.001, which seems implausibly low for distributions that appear visually similar.
 - a. We thank the reviewer for highlighting this. The explanation for p-value calculation was provided in the Supplementary Material, but we acknowledge that this was not clearly referenced in the main text. The manuscript has now been revised to explicitly direct readers to the relevant section in the Supplementary Material.
 - b. Regarding Figure S1a, while the distributions may appear visually similar, subtle differences can be observed. For instance, the mean value for "100 vs. 90" is slightly higher and exhibits a tighter distribution compared to "100 vs. 100". Although these differences seem minor, the large sample size contributes to the small p-value. Specifically, there are 30 simulations of ROIs with 90 points per cluster and 30 simulations of ROIs with 100 points per cluster, resulting in 900 pairwise comparisons. Given this high statistical power, even small differences yield low p-values.
9. Figure 3 mentions similarity to the Jurkat TIGIT dataset, which, as previously noted, appears excessively similar to multiple other datasets. This raises concerns about the accuracy of the similarity metric.
 - a. We thank the reviewer for highlighting this concern. As noted in our response to the previous comment regarding the Jurkat TIGIT dataset, the similarity metric can be influenced by differences in localisation density. The manuscript

now explicitly discusses this limitation and emphasises the importance of using the thinned similarity score for density-independent comparisons.

Reviewer 3

1. The methodologies used are generally poorly described in the manuscript. For example, the description of the dissimilarity algorithm is extremely short and hard to understand. Similarly, there are no details provided about how the simulations are performed. It would be beneficial to the reader to have a more in-depth description of these methods.
 - a. We thank the reviewer for their feedback and agree that a more detailed description of the methodologies would enhance clarity. In response, we have now expanded the descriptions of the dissimilarity algorithm and simulation methods in the supplementary material to provide a more comprehensive explanation for the reader.
2. Along these lines, the database does thinning to fairly compare data sets with different densities (by for example different measurement conditions). How is this thinning performed? Just a random selection of points? And what happens if the data is sparser than 30 localizations per μm^2 . It is also unclear how this comparison works, as many of the databank entries have multiple files for the entire dataset. Are these first combined into the full structure, or are they analyzed separately?
 - a. We thank the reviewer for their helpful comment. In response, we have now included a more detailed description of the thinning process in the supplementary material, outlining how points are selected, how comparisons are made across multiple files within a dataset, and what occurs when data is sparser than the target thinning density.
 - b. To improve the accuracy and robustness of the similarity algorithm, we have adjusted the default thinning density from 30 to 100 localisations/ μm^2 . This decision reflects a balance between maintaining sufficient spatial information for meaningful comparison and avoiding algorithmic underperformance at excessively low densities. The effect of thinning data on the similarity algorithm has now been included as an extra figure in the supplementary material (figure S5). Our analysis showed that the algorithm performs more reliably at this higher density threshold, while nearly all datasets in the database (42/43 datasets) contain localisation files with densities that still meet or exceed this value. As such, this updated threshold improves performance without significantly limiting the inclusion of available data.
 - c. For comparisons involving datasets with multiple files, each file is split into $3 \times 3 \mu\text{m}^2$ ROIs, and thinning is performed on each ROI across all files to a target density of 100 localisations/ μm^2 via random subsampling of points. If an ROI has a lower density than this threshold, no thinning is applied and that particular ROI is removed from analysis. ROIs from all files within a dataset are pooled and treated equivalently — they are not combined into a single structure, but rather analysed as independent ROIs. The similarity algorithm is then applied to compare all ROIs from one dataset to those from another, and the resulting similarity scores (calculated at the ROI level) are averaged, with the mean and standard deviation reported to summarise the comparison.
 - d. These clarifications and parameter changes have been incorporated into the supplementary materials to enhance transparency and reproducibility.
3. A link on the databank website that points to an “algorithm explanation/help” page would also be useful, so users can double check what these scores mean.

- a. In response, we have now included an algorithm explanation/help page, which can be accessed when viewing the ‘Similarity Search Results’ for a given dataset. This provides users with a clear reference to understand the meaning of the similarity scores.
4. 20+ of the datasets (out of 33) are “most similar” with the Jurkat_TIGIT data set, even the “actin HeLa Phalloidin 488” and “actin HeLa Phalloidin 647” point to this dataset, whereas one would expect it to point to each other or another actin data set. Did something go wrong in the selection there?
 - a. We thank reviewer 3 for highlighting this issue and reviewer 2 raised a similar concern. A known limitation of the algorithm is that similarity scores can be skewed by localisation density with lower-density datasets tending to yield lower scores. Since the 'Jurkat_TIGIT' dataset has a very low average density, it appears as the most similar dataset to many others when using the unthinned similarity score.
 - b. To mitigate this bias, the thinned similarity score is computed by equalising localisation densities before applying the algorithm, providing a density-independent comparison. When using thinned scores, 'Jurkat_TIGIT' is no longer the most similar dataset for the majority of cases.
 - c. To ensure clarity, we have now included guidance in the help section, recommending the use of thinned similarity scores when comparing datasets with significantly different densities.
5. While the databank contains some metadata parameters, it does not go far enough to ensure that the data uploaded is of high quality. Points such as signal to noise ratio of the raw data, background, whether the number of fluorophores active per frame is below the sparseness required for the fitting algorithms used, imaging completeness, under-sampling all affect the final image quality. It would be much more useful if the uploaded data was screened and given a quality score. In the absence of this, the usefulness of the curated data and the comparisons is unclear. This limitation should at the very least be discussed.
 - a. While we cannot guarantee the absolute quality of all uploaded data, we have implemented several metrics and mechanisms to help users assess data quality.
 - b. Localisation density is already included as a metric to assist in evaluating dataset quality. Additionally, upon upload, users must specify whether the data has undergone drift and blink correction. In response to the reviewer’s suggestion, the database now calculates and displays the average localisation precision for each dataset, providing an additional quality assessment tool.
 - c. To further enhance transparency, we have introduced a field where users can provide a link to their raw data, allowing independent evaluation. Users are also encouraged to include contact details, facilitating discussions and potential collaborations between researchers.
 - d. While the database itself cannot enforce data quality standards, it provides the tools and transparency needed for researchers to assess and compare datasets effectively. This discussion has now been incorporated into the manuscript.
6. It appears that the viewer can only see the small field of view per file. Some databank entries are so split up (e.g., limited frames) that there is no structure visible. This is particularly true for many of the Lck mEos3.3 Jukart on anti-CD90 datasets, where many fields of view contain just a few, sparse, random localizations. This makes it hard to judge something quickly without first having to download all the files. It would be nice to be able to see the full, unsplit field of view.

- a. We thank the reviewer for their comment and appreciate their concern regarding the small fields of view. We agree that having access to the full, unsplit field of view improves data interpretation and usability.
 - b. To address this, we now require users to upload whole fields of view along with a polygon overlay, ensuring that the full dataset can be visualised easily. Additionally, where possible, we have replaced existing datasets with their full fields of view to enhance clarity and accessibility.
7. It is unclear that splitting the fields to $3 \times 3 \mu\text{m}^2$ to do the “dissimilarity” comparison is useful as it assumes that the protein is uniformly distributed throughout the cell. For proteins that show biased localization to specific organelles or sub-cellular compartments (e.g. peri-nuclear region, nucleus etc...), this information will be lost when the fields are randomly split into small fields of view. In addition, proteins that look the same at nanoscale level may not necessarily be the same when cell-scale or mesoscale information is considered and hence, additional metrics in addition to the “dissimilarity metric” are likely necessary to truly compare different datasets. These points must be considered and addressed to make the resource more broadly useful than just looking at clusters of membrane proteins. Otherwise, the “spatial nano-omics” label is misleading.
 - a. The focus of this database is on the nanoscale architecture of proteins within cells, as this is a key strength of SMLM. While we acknowledge that sub-cellular organisation plays a crucial role in protein localisation, SMLM is particularly suited for analysing spatial patterns at the nanoscale, rather than mesoscale or whole-cell distribution.
 - b. That said, we recognise that proteins often exhibit heterogeneous spatial organisation within cells. To address this, we provide a whole-cell field of view image, which allows users to observe broader spatial context. Additionally, we have now incorporated a self-similarity histogram, which can reveal the presence of distinct subpopulations within a dataset. If a dataset contains two separate localisation patterns, this will be reflected in the histogram as two distinct peaks.
 - c. To further support this approach, we have validated the self-similarity histogram using simulated data, and the results are now included in the supplementary material.

Reviewer comments

- Figure 1 still only mentions .csv files (figure & legend). This should be updated.
 - We thank the reviewer for spotting this, this has now been rectified
- When thinning is mentioned in the text (e.g., line 90), a reference should be made to the supplementary information where the information on the process can be found.
 - We thank the reviewer for their comment, a reference to where more detail on the process can be found is made (originally in the supplementary section but now in the methods section in the main manuscript)
- The last statement of the SI that mentions that the thinning dissimilarity value is preferred for data with different densities should also be mentioned in the main text as this is very important information.
 - We thank the reviewer for this comment, the main text has now been altered to include this
- Some dissimilarity scores are below 0 (e.g., figure S6), which does not seem possible according to the definition that is provided in the SI? $c(\alpha)$ is a number that will always be positive, J is always positive, and D_{mn} contains the absolute value, so it is also positive. Is this an artifact from the violin plot representation?
 - We thank the reviewer for noticing this. This is an artifact from the violin plot representation. The violin plots estimate and illustrate data distributions using the kernel density estimation. This creates the data with ‘tails’, where the minimum tail does not represent the minimum data point. We can confirm that all similarity scores are positive.
- The following resource cannot be accessed without an account: For more details of how simulations are generated, see <https://gitlab.bham.ac.uk/shirgils/nano-org-similarity-scoring> (lines 90-91 in the SI). This should be made publicly available as it contains pertinent information to how the simulations are performed.
 - We thank the reviewer for spotting this. This has now been made public
- Line 109 of the SI does not appear to be shown correctly.
 - We thank the reviewer for highlighting this formatting error. It has not been fixed.